# Hemodynamic Instability during Surgery for Pheochromocytoma: A Retrospective Cohort Analysis

**DOI:** 10.3390/jcm11247471

**Published:** 2022-12-16

**Authors:** Moritz Senne, Doerte Wichmann, Pascal Pindur, Christian Grasshoff, Sven Mueller

**Affiliations:** 1Department for Visceral, General and Transplant Surgery, Tübingen University Hospital, 72076 Tübingen, Germany; 2Department of Anaesthesiology and Intensive Care Medicine, Tübingen University Hospital, 72076 Tübingen, Germany; 3Department of General and Visceral Surgery, Helios Clinics Gifhorn, 38518 Gifhorn, Germany

**Keywords:** pheochromocytoma, hemodynamic instability, adrenalectomy

## Abstract

Background: Perioperative hemodynamic instability is one of the most common adverse events in patients undergoing adrenalectomy for pheochromocytoma. The aim of this study was to analyze the impact of perioperative severe hemodynamic instability. Methods: We present a retrospective, single-center analysis in a major tertiary hospital of all consecutive patients undergoing elective adrenalectomy from 2005 to 2019 for pheochromocytoma. Severe perioperative hypertension and hypotension were evaluated, defined as changes in blood pressure larger than 30% of the preoperative patient-specific mean arterial pressure (MAP). Results: Unilateral adrenalectomy was performed in 67 patients. Intraoperative episodes of hemodynamic instability occurred in 97% of all patients (*n* = 65), severe hypertension occurred in 24 patients (36%), and severe hypotensive episodes occurred in 62 patients (93%). Patients with more than five severe hypotensive episodes (*n* = 29) received higher preoperative alpha-adrenergic blockades (phenoxybenzamine 51 ± 50 mg d^−1^ vs. 29 ± 27 mg d^−1^; *p* = 0.023) and had a longer mean ICU stay (39.6 ± 41.5 h vs. 20.6 ± 19.1 h, *p* = 0.015). Conclusion: Intraoperative hypotensive, rather than hypertensive, episodes occurred during adrenalectomy. The occurrence of more than five hypotensive episodes correlated well with a significantly longer hospital stay and ICU time.

## 1. Introduction

Pheochromocytoma is a rarely diagnosed tumor with an incidence of one to three per one million habitants [1]. Because of their neuroectodermal origin, pulsatile catecholamine secretion leads to several symptoms, such as hypertensive blood pressure and paroxysmal palpitations, anxiety, sweating, and headaches [2,3]. Cases of lethal outcomes have been described due to stroke and myocardial infarction following severe hypertension in undiagnosed settings [4]. The laparoscopic removal of pheochromocytoma after medicinal alpha-adrenergic blocking to prevent an intraoperative hypertensive crisis is the most commonly used treatment [2,5]. Nonselective alpha-adrenergic blocking with phenoxybenzamine is also still widely used [6]. Alternatively, doxazosin, a selective alpha-locker, is increasingly used in some centers [7].

During surgery, hemodynamic instability, in terms of hypertensive crisis and tachycardia due to the manipulation of the tumors and severe hypotension after vessel interruption, are the most frequently described intraoperative complications [8,9,10,11]. It is known that severe intraoperative hypotension in noncardiac surgery can lead to complications like acute kidney injury, stroke, or myocardial infarction [12,13]. The aim of this study was to assess the occurrence of intraoperative episodes of hyper- and hypotension during surgery for pheochromocytoma and their correlation with postoperative outcomes.

## 2. Materials and Methods

### 2.1. Study Setting

This retrospective study was approved by the local ethics review board (069/2018BO2). Using our in-house patient data management system (PDMS; SAP, Walldorf, Germany), we analyzed all anesthetic digital records from adults who underwent primary unilateral adrenalectomy due to pheochromocytoma at Tübingen University Hospital between 2005 and 2019.

### 2.2. Patients

All patients (≥18 years) with pheochromocytoma and elective adrenalectomy were included. Pheochromocytoma was diagnosed either by preoperative elevated urinary or plasma-free catecholamines, metanephrines [14], and/or positive histology [1]. We excluded all cases of other disease entities, emergency surgery, or preoperative ICU stays, as well as patients with inconsistent diagnoses of pheochromocytoma or those missing primary outcome and follow-up data.

### 2.3. Anesthesiologic Management

After arrival in the anesthesia induction room, an intravenous infusion was implemented, and the noninvasive monitoring of vital parameters was applied. General anesthesia was induced by intravenous injection of sufentanil (Hameln Pharma GmbH, Hameln, Germany) 0.4–0.6 µg kg^−1^, 2–3 mg kg^−1^ propofol (Fresenius Kabi Deutschlang GmbH, Bad Homburg, Germany), and 0.5–0.9 mg kg^−1^ rocuronium (Inresa Arzneimittel GmbH, Freiburg, Germany). After the intubation of the trachea, a radial arterial line and a three-lumen central-venous catheter were inserted. General anesthesia was maintained with sevoflurane (AbbVie Deutschland GmbH, Ludwigsburg, Germany) (0.7–1.0 MAC) and the intravenous boli of sufentanil and rocuronium at the discretion of the anesthesiologist. Norepinephrine (Cheplapharm Arzneimittel GmbH, Greifswald, Germany) and glycerol trinitrate (Carinopharm GmbH, Elme, Germany) were used to stabilize the blood pressure. After the end of the surgery, the patients were promptly extubated.

### 2.4. Outcome Measures and Variables

The preoperative patient-specific average MAP was calculated by averaging three to five MAP values measured before the start of general anesthesia. The intraoperative MAP was digitally recorded every minute using an arterial catheter. The primary outcome was the occurrence of hemodynamic instability, defined as a 30% deviation of the intraoperative MAP from the preoperatively calculated patient-specific average MAP. Furthermore, deviations were assessed for the number of episodes [15]. An episode was defined by the course of MAP exceeding the ±30% [12,13] limit of the preoperative patient-specific average MAP for at least one minute or longer. The population was divided into two groups based on the mean value of the hypotensive episodes. Pre- and perioperative treatments (preexisting alpha-adrenergic blockade, alpha-adrenergic blockade daily dosage (mg phenoxybenzamine), intraoperatively applied catecholamines, intraoperatively administered fluids, and glycerol trinitrate use) were analyzed, and complications were graded according to the Clavien–Dindo classification [16]. Furthermore, the following patient characteristics were assessed: age, gender, body size (BMI), final histopathologic diagnosis, presence of preoperative urinary catecholamines elevated more than sixfold according to Namekawa et al. [17], previous abdominal surgery, largest pheochromocytoma diameter in preoperative CT scan, pheochromocytoma site, type of surgery (laparoscopy vs. open), surgery time (minutes), date of surgery, postoperative ICU stay (hours), pre- and postoperative serum creatinine (mg dL^−1^), urinary output in the first 24 h after surgery (mL d^−1^), pre- and postoperative (24 h) hemoglobin concentration (g dL^−1^), platelet count (per µL), preexisting comorbidities (cardiovascular disease, arterial hypertension, diabetes mellitus), redo surgery, and 30-day hospital readmission.

### 2.5. Data Management

Data were reported according to the STROBE guidelines for retrospective cohorts [18]. Statistical analysis was performed using SPSS Statistics 22 for Windows (IBM Corporation, Armonk, NY, USA). Descriptive statistics were expressed as mean [standard deviation] or median [range] as appropriate. The Mann–Whitney U and the Kruskal–Wallis test were used to compare continuous variables, while the Chi-square and Fisher’s exact tests were used for the comparison of discrete variables. A *p*-value less than 0.05 was considered to be statistically significant. The factors of age, sex, preoperative comorbidities (diabetes, renal insufficiency, coronary heart disease), pheochromocytoma size, preoperative catecholamine levels, preoperative patient-specific MAP, surgery time or type, and preoperatively described hypertensive crisis were analyzed in a univariate logistic regression analysis if they were associated with more than five severe intraoperative hypotensive episodes.

## 3. Results

During the study period, a total of 345 patients underwent adrenalectomy. We excluded two patients with pheochromocytoma, one because of missing intraoperative and postoperative blood pressure data and the other because of required preoperative ICU treatment. In 276 patients, no pheochromocytoma was a reason for resection. Finally, 67 patients with pheochromocytoma were included and analyzed. Patient characteristics are displayed in Table 1. Intraoperative hemodynamic instability occurred in 65 patients. Severe episodes of hypotension occurred in 62 patients, and episodes of severe hypertension occurred in 24 patients. The 24 patients with severe hypertension had a mean of 2.1 ± 1.6 episodes, and the 62 patients with severe hypotension had a mean of 5.4 ± 2.8 episodes.

A cutoff value of more than five episodes of hemodynamic instability was found to correlate to a significantly longer ICU (39.6 ± 41.5 h vs. 20.6 ± 19.1 h, *p* = 0.015) and hospital stay (8 d [4–25] vs. 5 d [2–16], *p* = 0.011). No correlation was found between the number of episodes of hemodynamic instability and postoperative complications. Detailed results of episodes and duration of hemodynamic instability are displayed in Table 2. Patients with more than five severe intraoperative hypotensive episodes were identified to have a significantly higher preoperative dosage of alpha-adrenergic blockade (51 ± 50 mg d^−1^ vs. 29 ± 27 mg d^−1^ phenoxybenzamine; *p* = 0.023) and a higher preoperative patient-specific average MAP (112.2 ± 13.2 mmHg vs. 99.2 ± 17.5 mmHg; *p*= 0.0014).

No associations were found between age, sex, preoperative comorbidities (diabetes, renal insufficiency, coronary heart disease), pheochromocytoma size, preoperative catecholamine levels, preoperative patient-specific MAP, surgery time or type, comorbidity, preoperatively described hypertensive crisis, and severe intraoperative hypotensive episodes. To exclude a change in perioperative treatment over time, two different time periods (February 2005–2013 and March 2013–2019) were analyzed. No differences were found between these two periods regarding the occurrence of hemodynamic instability, surgery time, intraoperative norepinephrine, and glycerol trinitrate use.

## 4. Discussion

This study showed that severe hemodynamic instability during surgery for pheochromocytoma was a frequent event, with almost all patients being exposed to hypotensive episodes for a relevant time, while hypertensive episodes occurred less frequently in only one-third of the patients. Specifically, patients with more than five episodes of severe hypotension needed more perioperative fluid administration and showed a significantly longer ICU and hospital stay, while severe hypertensive episodes had no impact. It is well known that patients with larger and more hormonally active tumors are at higher risk for extreme hypertensive episodes during surgery [9,19]. A work by Tauzin-Fin et al. showed that hypertensive peaks during resection for pheochromocytoma occurred mainly during the creation of the pneumoperitoneum and tumor manipulation, while hypotension occurred mainly during and after tumor resection [20]. Even though a number of studies have tried to identify risk factors for intraoperative hemodynamic instability, individual perioperative courses are often unpredictable, posing a challenge to the anesthesiologist [21,22]. Chen et al. [23] reported that the type of surgery had an impact on hemodynamic instability. However, in our analysis, we did not find a difference between open trans-abdominal and minimally invasive surgeries. Interestingly, in our study, episodes of hypotension were observed, not only after resection but throughout the whole surgical procedure. Therefore, close perioperative hemodynamic monitoring is necessary to protect the patient from possible complications. In general, intraoperative hemodynamic monitoring with continuous arterial blood pressure measurement is recommended for pheochromocytoma surgery to detect rapid blood pressure changes early [24]. The impact of severe perioperative hypotension during pheochromocytoma surgery is described far less often.

Prolonged intraoperative hypotension is shown to be associated with increased organ failure, such as perioperative acute kidney injury within the first few days [25], myocardial events with elevated troponin [12], postoperative stroke [15], and overall 30-day mortality after noncardiac surgery [26]. However, intraoperative hemodynamic instability is not accurately defined. Some authors have defined intraoperative hypotension as a time-related drop below a previously defined value of MAP [26,27], while others have used relative changes related to patient-specific preoperative MAP [12,13]. There is no consensus on what definition should be used to describe relevant hypotension or hypertension during surgery [28]. Using absolute values to describe hypo- and hypertension might be easier [13], but some authors have propagated a relative drop in the MAP of more than 30% compared to the preoperative patient-specific average MAP [15,25] to consider patients’ individual normal blood pressure. Therefore, we decided to use deviations from the individual normal pressure as a marker. Hypotension after resection for pheochromocytoma is mainly described in the early postoperative course and may be the reason for postoperative intensive care treatment [17,19]. This is explained by persisting circulating antihypertensive medication, the downregulation of adrenergic receptors, and the reversal of chronic vasospasm [17]. In general practice, perioperative hypotension is mostly treated reactively after low blood pressures have already been recorded [29]. In addition, severe intraoperative hypotension might be a result of a higher preoperative dosage of phenoxybenzamine, a nonselective and long-acting alpha-adrenergic antagonist [30]. Patients with more than five episodes of severe hypotension had a significantly higher preoperative dosage of phenoxybenzamine than patients with fewer episodes in our series. A preoperative alpha receptor blockade has historically been assumed to prevent cardiovascular morbidity and mortality due to excess hypertensive periods [31]. Due to this fact, preoperative alpha-blockage is widely used to reduce intraoperative hypertensive crises and is recommended in several guidelines [6,32]. However, routine preoperative administration remains doubtful, as no controlled trials exist on this issue. Several uncontrolled trials reported no relevant effects of preoperative alpha receptor blockades on morbidity after pheochromocytoma surgery [33,34]. Buitenwerf et al. demonstrated, in a prospective randomized trial comparing phenoxybenzamine and doxazosin, an advantage for phenoxybenzamine in reducing intraoperative hemodynamic instability, but they did so without showing a difference in postoperative outcome and with no effect on the duration of blood pressure outside the target range during surgery [7]. A multicenter retrospective survey by Groeben et al. even showed that cases without preoperative alpha receptor blockades had fewer cardiovascular events in the peri- and postoperative course than patients with routine alpha receptor blockades [35]. Furthermore, the incidence of severe intraoperative hypertensive periods was seen to have no effect when comparing patients treated with and without preoperative alpha receptor blockades [34]. In a systematic review, Schimmack et al. compared several proceedings with selective or nonselective preoperative alpha receptor blockades. The authors did not describe an impact on hemodynamic stability [36]. Urapidil, as a competitive selective short-acting α1 blockade, was investigated by Tauzin-Fin et al. in a retrospective analysis of a cohort of 79 cases. In contrast to Schimmack et al., they evaluated urapidil to potentially result in lower rates of hemodynamic instability peaks [20]. In this context, a competitive selective short-acting α1 blockade would be an alternative to the more commonly used phenoxybenzamine.

The limitations of this study included its retrospective design, the limited number of analyzed patients, and lacking a consensus on the definition of intraoperative hemodynamic instability. No association was found between hemodynamic instability (either hypotensive or hypertensive episodes) and complication grade. As complications ranged around 10% after surgery for pheochromocytoma, manifold larger sample sizes in a trial are needed to detect differences. There is evidence for both the absolute value of MAP (<60–70 mmHg) and the relative deviation of patients’ individual MAP (−30%) for postoperative myocardial complications. Randomized prospective trials are needed to decide which parameter should be used.

## 5. Conclusions

In summary, this retrospective study showed that patients undergoing adrenalectomy for pheochromocytoma displayed more and longer intraoperative episodes of severe hypotension than hypertension, resulting in a greater need for intraoperative crystalloid fluid substitution and longer ICU and hospital stays. The general recommendation of the preoperative use of alpha-adrenergic receptor blockades demands further investigation, considering short-acting alpha-blockade options.

## Figures and Tables

**Table 1 jcm-11-07471-t001:** Patient characteristics.

Items	Total	Severe Hypertension	Severe Hypotension
Patients (*n*)	67	24	62
Median age (years)	54 [25–87]	57 [25–81]	54 [25–87]
Gender, female (*n*)	37	16	34
Mean Body Mass Index (kg m^−2^)	26 ± 6.8	27 ± 8	26 ± 6.1
Mean maximum radiologic diameter (cm)	4 ± 1.9	5.1 ± 1.9	4.3 ± 1.9
Right Localisation (*n*)	35	10	34
Median surgery time (min)	113 [35–291]	115 [36–205]	116 [35–291]
Preoperative employed alpha blockade (*n*)	58	19	55
Mean dosage alpha blockade ^a^ (mg d^−1^)	39 ± 40	44 ± 55	40 ± 40
Preexisting hypertensive crisis (>180/120 mmHg) (*n*)	21	7	20
preoperative patient-specific average MAP (mmHg)	108 ± 17	101 ± 21	108 ± 16
Preoperative > 6-fold elevated catecholamines (*n*)	16	5	16
Comorbidity (*n*)			
Diabetes	22	8	20
Coronary Heart Disease	11	4	11
Renal insufficiency	9	3	7
Type of surgery (*n*)			
Minimal invasive	52	19	47
Open	15	5	15
Mean duration of episodes (min)		11.8 ± 14.8	62 ± 47.7
Mean episodes (*n*)		2.1 ± 1.6	5.4 ± 2.8

^a^ Phenoxybenzamine; MAP, mean arterial pressure.

**Table 2 jcm-11-07471-t002:** Details of preoperative average MAP, along with episodes and duration of hemodynamic instability.

Number of Episodes of Severe Hypotension	≤5	>5	*p*-Values
Patients (*n*)	38	29	
Mean duration of hemodynamic instability (min)	39.6 ± 51.3	66.6 ± 41.6	0.017
Preoperative patient-specific average MAP (mmHg)	99.2 ± 17.5	112.2 ± 13.2	0.0014
Mean intraoperative norepinephrine (µg kg^−1^ min^−1^)	0.13 ± 0.16	0.19 ± 0.2	0.177
Episodes of severe hypertension	0.87 ± 1.6	0.62 ± 0.98	0.471
Mean intraoperative glycerol nitrate (µg kg^−1^ min^−1^)	0.09 ± 0.26	0.26 ± 0.52	0.08
Mean intraoperative crystalloids (mL)	1922 ± 1256	2707 ± 1606	0.032
Mean 24 h crystalloids (mL)	3068 ± 2190	3809 ± 1800	0.176
Preoperative dosage alpha blockade (mg d^−1^ Phenoxybenzamine)	29 ± 27	51 ± 50	0.023
Median ICU time (h)	20.6 ± 19.1	39.6 ± 41.5	0.015
Median length of hospital stay (d)	5 [2–16]	8 [4–25]	0.011
Complication Clavien–Dindo (13) (*n*)			0.311
≤II	9	7	
≥III	1	4	
Any preoperative antihypertensive drug intake (except phenoxybenzamine)	24	21	0.447

MAP = mean arterial pressure; ICU = intensive care unit.

## Data Availability

The datasets generated and/or analyzed during the current study are not publicly available due to institutional privacy policy but are available from the corresponding author upon reasonable request.

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
