# Peer review of "Hemodynamic Instability during Surgery for Pheochromocytoma: A Retrospective Cohort Analysis"

_jcm, 2022, doi:10.3390/jcm11247471_

Round 1
Reviewer 1 Report
This is an intriguing manuscript and I think provides us with legitimate questions about periopeative management for pheochromocytomas. I would like the authors to expand on the relationship between intraoperative hypertension and icu length of stay. I understand a larger study would help prove this but can the authors more definitely make this link?
Author Response
Thank you for your comment. In our retrospective study we had no relationship between intraoperative severe hypertension and longer ICU stay. Patients with more than 5 episodes of severe hypotension had a longer ICU stay, but we cannot demonstrate a definite difference in the reasons for intensive care therapy between the groups, so we did not include this detail. In addition to the risks of intraoperative hypertensive crisis and the resulting cardiovascular stress, we have now highlighted in the discussion the option of postoperative prolonged hypotension, which has been discussed in other studies.
Reviewer 2 Report
The manuscript by Senne et al. is interesting, well-written and deals with a subject where more knowledge is needed, i.e., the hemodynamic instability during adrenalectomy for pheochromocytomas. I have the following comments that needs to be addressed:
1. Line 18: Since the total number of patients included where less than 100, please do not use a decimal in percentages. This should be changed throughout the manuscript.
2. Line 29-30: Hypertension is common, however, flush may not be so common. It is more that the patients may feel hot in around a quarter (see Falhammar H et al Endocrine Connections 2018). I think what you are trying to say is paroxysmal symptoms. Please change and maybe also cite the suggested reference.
3. Line 33: The reference is quite old, please add a modern reference as well to reflect modern practice, e.g., Calissendorff J et al Cancers 2022.
4. Line 34: Doxazosin, a selective alpha-blocker, is used very often nowadays, it could be mentioned as well.
5. Line 51-52: Metanephrines in plasma or urine are the recommended biochemical test for diagnosing pheochromocytoma. Were these not used?
6. Line 69: Please write 30% in numbers and not letters.
7. Table 1: I note that not all patients had had preoperative alpha-blockage, how come?
8. Line 187 and forward: The RCT study by Buitenwerf E JCEM 2020 could be mentioned briefly here.
Author Response
Thank you for reading and commenting on our manuscript. In the following we have processed your comments:
1.1: Thank you for your comment. We changed the percentages and rounded them accordingly.
2.1: Thank you for your comment. We specified the symptoms and added the suggested reference.
3.1: Thank you for your comment. We updated the reference.
4.1: Thank you for your remark. We added a note that doxazosin is used alternatively in some centers.
5.1: Both catecholamines and metanephrines were used for diagnosis. We have supplemented this.6.1: Thank you for your comment, we changed that.
7.1: Thank you for this remark. The reason for this cannot always be traced in the patient data.
8.1: Thank you for your comment and the suggested literature. We inserted the results of Buitenwerfen et. al and relate the results to the already cited studies.